# The Nutritional Status of Adult Antiretroviral Therapy Recipients with a Recent HIV Diagnosis; A Cross-Sectional Study in Primary Health Facilities in Gauteng, South Africa

**DOI:** 10.3390/healthcare8030290

**Published:** 2020-08-24

**Authors:** Khabo Mahlangu, Perpetua Modjadji, Sphiwe Madiba

**Affiliations:** Department of Public Health, School of Health Care Sciences, Sefako Makgatho Health Sciences University, Pretoria 0001, South Africa; elizabethmahlangu58@gmail.com (K.M.); sphiwe.madiba@smu.ac.za (S.M.)

**Keywords:** South Africa, antiretroviral therapy, adults, underweight, overweight, obesity

## Abstract

The study determined the nutritional status of adult antiretroviral therapy (ART) recipients, and investigated the association between the duration on ART and the nutritional status. This study was based in primary health facilities in Gauteng, South Africa. The data collected included sociodemographic variables; the duration of the treatment; and the body mass index (BMI), classified as undernutrition (<18.5 kg/m^2^), normal (18.5–24.9 kg/m^2^), or overweight/obesity (≥25 kg/m^2^). ART recipients (*n* = 480) had a mean age of 35 (± 8.4SD) years. All had taken ART for six months or more (range 6–48 months). The data were analyzed using STATA 13.0. The overall prevalence of overweight/obesity was 39%, it was higher in females (46%) than in males (30%), 26% were overweight, and 13% were obese. Underweight was 13%, and was higher in males (18%) than females (9%). Being overweight was more likely in those aged ≥35 years and those in smaller households. Being obese was less likely in males, in the employed, and in those with a higher income, but was more likely in those with a longer duration on ART. Abdominal obesity was high, but less likely in males. Interventions to prevent overweight/obesity should be integrated into routine HIV care, while at the same time addressing the burden of undernutrition among ART recipients.

## 1. Introduction 

The introduction of antiretroviral therapy (ART) has greatly altered the morbidity of populations affected by HIV [1,2]. ART contributes to the wellbeing of people living with HIV (PLHIV), and reduces the threat of the ongoing transmission of HIV, particularly to sexual partners and through the mother-to-child transmission of HIV [3]. Nonetheless, new challenges in the management of HIV/AIDS have emerged, and non-adherence to ART is the most significant challenge in developing and developed countries [3]. In developing and resource-limited communities, non-adherence is largely attributed to food insecurity [4,5,6,7,8]. Adequate nutrition is necessary for optimal treatment outcomes. Nutrition plays an important role in rebuilding worn-out tissues, it strengthens the immune system, and it delays the progression from HIV to AIDS-related diseases [6,9]. 

In resource-limited settings, many PLHIV on long-term ART lack adequate nutrition [8,10,11,12]. Malnutrition is a marker for a poor prognosis among PLHIV and is the most common cause of immunodeficiency worldwide [13,14]. Although South Africa is described as a food secure country, with regard to PLHIV, 29% were food insecure, while 9% were severely food insecure [15]. On the other hand, the prevalence and incidence of overweight and obesity among PLHIV has increased over the past two decades, and mirror, or in some instances exceed, that of the national population trends they reside in [2,16,17,18]. The prevalence of obesity is also attributed to improved survival and the introduction of new ART agents [16]. The current major concern is the vulnerability of PLHIV to a higher risk of cardiovascular diseases than the general population, due to complex interactions between the risk factors and the HIV infection itself [19,20,21].

While a positive HIV diagnosis was associated with severe underweight in the beginning of the HIV epidemic, weight loss and wasting is no longer an inevitable outcome of the progression of HIV [16]. Instead, overweight and obesity are now commonly observed in high proportions of ART recipients [22,23]. The shift in weight gain among PLHIV is considered a side-effect of ART regimens [24], and/or an immunological response or a reflection of an increased CD4 cell count [25]. There are limited studies and data on the nutritional status of ART recipients in South Africa. This is despite South Africa having the largest ART program in the world, as well as the association of ART with patterns of weight changes, the occurrence of metabolic dysfunction, and the increased risk of contracting other non-communicable diseases in PLHIV [26,27]. 

The purpose of this study was to determine the nutritional status of adult ART recipients, and to investigate any association between the duration of ART and the nutritional status of the study sample. Overweight/obesity threatens the wellbeing of PLHIV, and there is a need to monitor the weight of those on ART programs [15]. There is a need for the screening and monitoring of excessive weight gain in patients on ART in HIV clinics in order to prevent and manage the risk of developing metabolic syndrome [28]. This is especially crucial with the adoption of the Universal Test and Treat (UTT), a strategy that makes ART available to all HIV-infected persons, regardless of their CD4 count [29]. UTT exposes many overweight/obese individuals to ART [23]. This article is intended for general clinicians, nutritionists, dietitians, treatment teams of ART, HIV researchers, policy development, psychosocial interventionists, and social workers.

## 2. Material and Methods

### 2.1. Design and Setting of the Study

This was a cross-sectional, descriptive study conducted between February and April 2018 in health facilities in the Tshwane Health District. The district is located in the City of Tshwane Municipality, Gauteng, South Africa. The municipality comprises urban, rural, and peri-urban areas, with a total population of 2,708,702, according to the District Health information System (DHIS, 2011) (https://www.hst.org.za/publications/NonHST%20Publications/Gauteng-%20Tshwane%20District.pdf). The district is demarcated into seven sub-districts with 79 health facilities, which offer HIV services to adults and children through the nurse-initiated management of antiretroviral treatment (NIMART). The setting for this study was three community health centers (CHCs) in sub-district 1. They were selected because they had the largest number of clients receiving ART, and had been offering ART for a longer duration of time than the eight other clinics. Although two of the CHCs are located in urban areas, they also provide services to clients from informal settlements. One facility was located in a peri-urban area, but also offered services to rural clients from neighboring communities.

The population of the study consisted of HIV-positive adults aged between 18 and 49 years. The participants were eligible to participate in the study if they had been recipients of ART for six months or more. The study excluded individuals who were pregnant and those who had obvious symptoms of opportunistic infection, or those who had cognitive impairment due to HIV/AIDS infection or sub-mentality at the time of the study. A sample size of 432 was calculated using the Rao soft sample calculator [30], taking into consideration a total number of 14,000 HIV-positive adults receiving ART and HIV care in the three facilities, a 5% margin of error, a 50% response rate, and 95% confidence interval. In addition, a 10% non-response rate was anticipated, which gave the final sample size of 480. The sample was proportionally allocated to each of the selected facilities, and the participants were selected using a convenient sampling method. The selected facilities used in this study offer ART refill and follow-up on an appointment basis for PLHIV. The appointment schedule list was used to identify and select potential participants who met the inclusion criteria. To minimize selection bias, the data were collected on different clinic days and times over three months. 

### 2.2. Data Collection Instruments and Procedures

The study used a researcher-designed, standardized, pretested questionnaire to collect the data. The questionnaire (Appendix A), captured socio-demographic variables including age, gender, marital status, educational status, employment status, household income, and household size. In addition, data on the duration on ART were collected. The lead investigator (K.M.) and trained research assistants collected the data. The questionnaire was translated into the participants’ preferred languages, including IsiZulu and Setswana. One-day training for the research assistants in preparation for the fieldwork was done by the second author (P.M.), who supervised the fieldwork. The tool was pre-tested on 5% of the sample, was both researcher- and self-administered, and was completed in a private room provided by the staff of the clinic. Where the questionnaire was self-administered, the lead investigator checked it for completeness on a daily basis. 

Anthropometric measurements were carried out to determine the nutritional status of the study participants, using body mass index (BMI). All of the measurements were performed using standardized World Health Organization (WHO) methods [31]. The weight was measured in light clothes and without shoes, in kg, to the nearest 0.1, using a D-quip smart scale throughout the life course of the project, with a margin error between 0.1–0.2% of the actual weight. The height was measured in cm, to the nearest 0.1 cm, using a stadiometer, while the participants were standing upright in a Frankfurt position. All of the measurements were taken three times, and the average was recorded. The body weight (kg) and height (kg/m^2^) were used to calculate the BMI and to classify underweight, normal, overweight, and obesity in adults [31]. Undernutrition was indicated as BMI < 18.5 kg/m^2^, normal BMI was 18.5–24.9 kg/m^2^, overweight was indicated by a BMI ≥ 25 to 29.9 kg/m^2^, and obesity by BMI ≥ 30kg/m^2^. The waist circumference (WC) and hip circumference (HC) were measured in cm, to the nearest 0.1 cm, using a non-stretchable plastic tape measure. Central obesity was defined as WC ≥ 94 cm for males and ≥ 80 cm for females [32]. Abdominal obesity was classified as a waist/hip ratio (WHR) of > 0.85 for females and >1.00 for males, while a waist-to-height ratio (WHtR) of 0.5 was used for both sexes [33].

### 2.3. Data Analysis

STATA 13 (Stata Statistical Software: Release 13, StataCorp, College Station, TX, USA) was used for all of the statistical analyses. The skewness and kurtosis test for normality was used to determine the distribution of the variables. The variables with a skewed distribution were presented as median [interquartile range (IQR)]. The medians for weight, height BMI, WC, WHR, and WHtR were compared by sex using the Mann–Whitney U test, and the results were presented as median (IQR). The prevalence of overweight, obesity, underweight, and abdominal obesity were stratified by age, gender, and the duration on ART, and were compared using a Chi-square test. We used bivariate logistic regression analysis to determine the associations of overweight/obesity, abdominal obesity, and undernutrition with independent variables, by calculating the crude odds ratio (OR), 95% CI, and *p* values. All of the categorical variables associated with the outcome variables that had a *p* value < 0.20 in the bivariate analysis were entered into the multivariate logistic regression model using a forward stepwise regression. The crude odds ratio (OR) and adjusted odds ratio (AOR) with a confidence interval (CI) of 95% were determined, and *p*-values < 0.05 were considered significant.

### 2.4. Ethical Considerations

Ethical clearance was sought and granted by Sefako Makgatho Health Sciences University’s Research and Ethics Committee (SMUREC/H/217/2017: PG) and the Gauteng Department of Health. The participants provided written informed consent translated into IsiZulu and Setswana, before completing the questionnaire.

## 3. Results

### 3.1. Sociodemographic Characteristics of Adult ART Recipients

In total, 480 adult ART recipients participated in the study. Over half (56%) of them were females. The mean age was 35 (±8.4 SD) years, the range was 18–49 years, and 55% were 35 years old or older. The majority (65%) were single, 50% were household heads, 61% lived in households with one to four members, 63% were unemployed, 77% were beneficiaries of social grants, 35% came from households with no income, and 48% came from households with a monthly income between $55.22 and $265.39. Concerning educational status, the majority (81%) had a high literacy level (i.e., completed Grade 12 and had tertiary education). Only 19% had limited literacy (i.e., primary and secondary education) (Table 1).

### 3.2. Nutritional Status of Adult ART Recipients

Table 2 presents the nutritional status of the participants. The results show that females had a higher mean weight (67 kg) than males (64 kg; *p* = 0.062). The median BMI (25.4 ± 6.4 kg/m^2^) of females was significantly higher compared with the median BMI (24.2 ± 6.4 kg/m^2^) of males (*p* ≤ 0.0001). The majority (48%) had normal BMI, 13% were underweight, 26% were overweight, and 13% were obese. Slightly more males (18%) than females (9%) were underweight, but slightly more females (27%) than males (25%) were overweight, and with regard to obesity, the prevalence was significantly higher for women (19%) than men (5%; *p* ≤ 0.0001). The overall overweight/obesity prevalence was high (39%), and significantly more females (46%) were overweight/obese than males (30%; *p* ≤ 0.0001). Abdominal obesity was prevalent, as indicated by WC (45%), WHR (38%), and WHtR (75%). Females had a higher abdominal obesity by WC (58%), WHR (47%), and WHtR (79%) than males (28%, 26%, and 70%, respectively). For the bivariate analysis, the results showed a significant association with gender. Males were less likely to have abdominal obesity (OR = 0.64; 95%CI: 0.42–0.97) and high WC (OR = 0.28; 95%CI: 0.19–0.41) than females. 

We further assessed the nutritional status of the participants by age (Table 3). Age was divided into two categories, namely < 35 years and ≥ 35 years. Slightly more participants aged ≥35 years were overweight (29% vs. 23%) and obese (15% vs. 10%) than those < 35 years. There was no difference in the prevalence of underweight and obesity (13% vs. 13%) in terms of age group. Similarly, the relation between abdominal obesity and age group was not significant (WC, 48% vs. 42%; WHR, 41% vs. 45%; and WHtR, 78% vs. 73%).

### 3.3. Nutritional Status and Duration on ART

The duration on ART ranged from 6 months to more than 2 years, and was divided into two sub-groups, namely <2 years and ≥2 years. We subsequently assessed the nutritional status of the participants in terms of their duration on ART, and 54% had been on ART for <2 years. The prevalence of overweight was higher (31%) in those with an ART duration of ≥ 2 years, compared with 21% for those with an ART duration of <2 years. However, underweight was more prevalent (15%) among those who had been on ART for <2 years than among those who had been on ART for ≥2 years (11%), and there was a statistical difference (*p* ≤ 0.0001). There was no statistical difference in the prevalence of abdominal obesity for WC (44% vs. 45%), WHR (37% vs. 38%), and WHtR (74% vs. 76%) by duration on ART (Table 4). 

### 3.4. Factors Associated with Nutritional Status among Adult ART Recipients

The bivariate regression analysis was performed to determine the association of overweight/obesity using independent variables (Table 5). In the bivariate analysis, underweight was associated with household income (OR = 0.42; 95 %CI: 0.23–0.74). Overweight was associated with age (OR = 1.55; 95% CI: 1.00–2.41) and household size (OR = 1.59; 95% CI: 1.01-2.53). Obesity was associated with the duration on ART (OR = 3.13; 95% CI: 1.72–5.71), gender (OR = 0.23; 95% CI: 0.11–0.47), employment (OR = 0.49; 95% CI: 0.26–0.91), and income (OR = 0.49; 95% CI: 0.26–0.93). 

To analyze the association between age and nutritional status, we stratified age into ≥35 and <35 years. For the bivariate analysis, age was significantly associated with overweight/obesity, but not associated with WC, WHR, and WHtR (Table 6). 

## 4. Discussion

In this study, we evaluated the nutritional status of PLHIV receiving ART in Gauteng, South Africa. The study found a high burden of malnutrition, with both undernutrition and over-nutrition being prevalent among adult ART recipients in urban and informal settings. The prevalence of overweight was 26%, obesity was 13%, undernutrition was 13%, and overall overweight/obesity was 39%. The double burden of undernutrition and overweight/obesity amongst PLHIV on ART has been reported in previous studies conducted in South Africa, Zimbabwe, Tanzania, and Ethiopia. Consistent with the rates in other studies conducted among PLHIV, the prevalence of overweight/obesity was significantly higher than underweight [34,35,36,37]. Current research shows that the prevalence of adult obesity now exceeds underweight in most countries of the world [38].

A previous study conducted in rural Limpopo among PLHIV and on ART reported a higher prevalence (36.4%) of overweight than the rates observed in the current study [39]. The difference could be attributed to the difference in the duration of ART; the Limpopo study selected participants who had been on ART for a longer duration (range: 1–121 months), whereas the current study was conducted among patients recently initiated on ART. This further explains the slightly higher prevalence of overweight (29%) and obesity (17%) reported in a 12-year retrospective study of HIV patients on ART in Botswana [40]. 

The current study found a high (39%) prevalence of overall overweight/obesity, and the prevalence was higher in women than in men. The association observed between gender and overweight/obesity has been previously documented [39,40,41]. The study established that age was also a factor associated with overweight. Participants ≥35 years were almost twice as likely to be overweight than those <35 years. Similarly, global data suggest that BMI among HIV-infected individuals on ART increasingly mirrors that observed among the general population [17].

We found that the prevalence of underweight (13%) was contrasted with the higher rates reported in studies in Ghana, Ethiopia, Senegal, and West Africa among PLHIV, where the rates ranged from 23–36%, [11,12,34,42], but were comparable with rates reported in Zimbabwe [37]. The prevalence of undernutrition in males was double (18%) than that of females (9%), although there was no significant association. Other studies have reported similar observations [37,39,43]. Among the HIV-infected people in sub-Saharan Africa (SSA), the high prevalence of undernutrition in men might be attributed to the delayed health-seeking behavior of men, who tend to begin HIV care later than women, when they would have lost more weight than women [43]. 

The current study further found that household income was a factor associated with undernutrition. ART recipients with a high household income were less likely to be undernourished than their counterparts. The findings are in line with those of other studies [11,43]. The socioeconomic status of households is a key to food access, and it has been indicated that PLHIV with adequate access are less likely to be undernourished compared with those with food insecurity. Evidence has shown that HIV/AIDS deepens food insecurity and affects the nutritional status of PLHIV, leading to weight loss [11].

This study found a higher prevalence of undernutrition in those who were on ART for a shorter duration (15%) than those who had been on ART for a longer duration (11%), although the association was not significant. This observation is supported by findings from other studies, in which those who had been on ART for a short duration were almost twice as undernourished as those who had been taking the ART for a longer duration [11,44]. The explanation for this observation could be that patients who are undernourished at the time of their enrolment on ART, who might have been suffering from a more advanced stage of the disease, become healthier and gain weight over time, with their recovery from opportunistic infections [41,45]. Research has indicated that patients in an advanced stage of the disease are more likely to be malnourished at the time of enrolment in HIV clinics [37,41]. However, evidence suggests that weight gain is common among HIV-positive individuals initiating ART [46].

The study further observed a high prevalence of abdominal obesity, as indicated by an elevated WHtR (75%), WHR (38%), and WC (45%). The study identified a statistical association between abdominal obesity and the female gender. The gender association reported in the current study has been observed in other studies in SSA [45]. This is of major public health concern, as patients who are overweight/obese and have abdominal obesity are predisposed to cardiovascular disease, diabetes, dyslipidemia, and metabolic syndrome [47].

The study found a significant association between the duration on ART and the nutritional status of ART recipients. Those who had been on ART for a longer duration were three times more likely to have developed obesity than those who had been on ART for a short duration. The increase observed in the current study in the prevalence of obesity after the commencement of ART is parallel to the rates reported in Côte d’Ivoire, Tanzania, and Nigeria, where large proportions of HIV-positive individuals were found to have progressed to overweight and obese states after being started on ART [25,41,48]. In cohorts of HIV-infected adults in the U.S., Canada, and Brazil, the prevalence and incidence of obesity after ART initiation continued to increase [49,50]. In these cohorts, 22% of individuals with a normal BMI became overweight and 18% of overweight individuals became obese within 3 years after ART initiation [50]. 

In line with other studies, this study found that various sociodemographic risk factors were associated with overweight/obesity [51,52]. These risk factors are similar to those observed in the general population [43,48]. In this study, gender, employment status, and income were associated with obesity in the univariate logistic analysis. The study found that men were less likely to be obese than women. It is imperative that preventive interventions for overweight and abdominal obesity should target HIV-infected individuals, especially women [45]. Household size was significantly associated with overweight. ART recipients in small household were almost twice as likely to be overweight than those in big households. 

In the multivariate logistic regression, age was significantly associated with the nutritional status of the participants as an independent indicator. Those aged 35 years and over were almost twice as likely to be overweight than their younger counterparts. Contrary to other findings [53], there was no association between abdominal obesity and age, but gender was an independent indicator for abdominal obesity. 

### Limitations

This study is not without its limitations. A major limitation is the use of a cross-sectional study design that limits inference. As such, we could not accurately show the cause and effect relationship of the duration on ART and weight gain. The study was conducted in one district in the City of Tshwane, and may not be representative of all ART recipients who are enrolled in HIV treatment, as the health facilities and participants were not randomly selected. Furthermore, the study could not collect or analyze the weight of the patient at enrolment (because of the quantity of missing and incomplete data) in order to draw comparisons and assess the trend in weight gain during ART care. We did not measure the participants’ dietary intake and practices in order to correlate that data with their overweight and obesity. However, our study is one of the few in South Africa to show the existence of a double burden of malnutrition among ART recipients in a peri-urban area. We believe that our study sheds light on the presence of overweight and obesity occurring, simultaneously with underweight, among ART recipients. 

## 5. Conclusions

The findings revealed that obesity/overweight was high and more prevalent than underweight among the ART recipients in this sample. The findings of this study have public health implications. First, there is a double burden of over-nutrition and undernutrition among ART recipients in health care facilities in the Tshwane District. Although the weight of the patients receiving ART is routinely measured, this is not linked to interventions to prevent the occurrence of overweight and obesity. Interventions to prevent overweight and obesity should be integrated into routine HIV care, while at the same time addressing the burden of undernutrition occurring alongside overweight/obesity among ART recipients. Secondly, the high prevalence of abdominal obesity increases the risk of cardiovascular diseases, with serious public health implications. Female gender and age were associated with the development of overweight, obesity, and abdominal obesity during ART. Large-scale longitudinal studies are needed to further assess the relationship between ART and weight gain, as well as the progression to overweight and obesity in patients started on ART in South Africa and other settings in SSA. 

## Figures and Tables

**Table 1 healthcare-08-00290-t001:** The sociodemographic characteristics of adult antiretroviral therapy (ART) recipients in health facilities in the Tshwane District (*n* = 480).

Variables	Categories	Frequency (*n*)	Percentage (%)
Age categories	<35 years	216	45
	≥35 years	264	55
Gender	Males	213	44
	Females	267	56
Marital status	Cohabiting	32	7
	Single	314	65
	Ever married	134	28
Level of education	Low literacy	91	19
	High literacy	389	81
Employment status	Unemployed	177	63
	Employed	303	37
Receiving a social grant	Yes	111	77
	No	369	22
Household size	1–4 members	294	61
	≥5 members	186	38
Household monthly income	No income	168	35
	$55.22	64	13
	$53.23–$265.39	166	35
	≥$265.44	150	17

**Table 2 healthcare-08-00290-t002:** Weight and nutritional status of adult ART recipients by gender.

Variables	All	Males (*n* = 213)	Females (*n* = 267)	*p*-Value
Weight (kg)	66 (40; 118)	64 (41; 110)	67 (40; 118)	0.062
Height (m)	1.65 (1.59; 1.72)	1.69 (1.62; 1.75)	1.63 (1.59; 1.68)	≤0.0001 *
BMI (kg/m^2^)	24.1 (14; 54.3)	22.6 (13.4; 44.9)	25.4 (14.7; 54.3)	≤0.0001 *
Normal	229 (48)	110 (52)	119 (45)	≤0.0001 *
Underweight	62 (13)	38 (18)	24 (9)	
Overweight	127 (26)	54 (25)	73 (27)	
Obesity	62 (13)	11 (5)	51 (19)	
WC	84.2 (75; 95)	85.43 (75; 96)	83.20 (75;91)	0.186
Normal	265 (55)	152 (72)	112 (42)	≤0.0001 *
Abdominal obesity	215 (45)	60 (28)	155 (58)	
WHR	0.84 (0.79; 0.89)	0.84 (0.79; 0.90)	0.84 (0.80; 0.88)	0.294
Normal	299 (62)	158 (74)	141 (53)	≤0.0001 *
Abdominal obesity	181 (38)	55 (26)	126 (47)	
WHtR	0.51 (0.45; 0.56)	0.51 (0.44; 0.58)	0.51 (0.46; 0.56)	0.33
Normal	120 (250	63 (30)	57 (21)	0.039 *
Abdominal obesity	360 (75)	150 (70)	210 (79)	

* Indicates significant differences

**Table 3 healthcare-08-00290-t003:** Nutritional status of adult ART recipients by age group.

Variables	All *n* (%)	Age (*n* = 216) <35 Years *n* (%)	Age (*n* = 264) ≥35 Years *n* (%)	*p*-Value
BMI				
Normal	229 (48)	115 (53)	114 (43)	0.127
Underweight	62 (13)	28 (13)	34 (13)	
Overweight	127 (26)	50 (23)	77 (29)	
Obesity	62 (13)	23 (10)	39 (15)	
WC				
Normal	265 (55)	113 (52)	152 (88)	0.249
Abdominal obesity	215 (45)	103 (48)	112 (42)	
WHR				
Normal	299 (62)	128 (59)	171 (65)	0.298
Abdominal obesity	181 (38)	88 (41)	93 (35)	
WHtR				
Normal	120 (250	48 (22)	72 (27)	0.204
Abdominal obesity	360 (75)	168 (78)	192 (73)	

**Table 4 healthcare-08-00290-t004:** Nutritional status of adult ART recipients by duration on ART.

Variables	All *n* (%)	ART < 2 Years (*n* = 219) *n* (%)	ART ≥ 2 Years (*n* = 261) *n* (%)	*p*-Value
BMI				
Normal	229 (48)	133 (51)	96 (44)	≤0.0001 *
Underweight	62 (13)	29 (11)	33 (15)	
Overweight	127 (26)	80 (31)	47 (21)	
Obesity	62 (13)	19 (7)	43 (20)	
WC				
Normal	265 (55)	122 (56)	143 (55)	0.840
Abdominal obesity	215 (45)	97 (44)	118 (45)	
WHR				
Normal	299 (62)	137 (63)	162 (62)	0.912
Abdominal obesity	181 (38)	82 (37)	99 (38)	
WHtR				
Normal	120 (250	58 (26)	62 (24)	0.492
Abdominal obesity	360 (75)	161 (74)	199 (76)	

* Indicates significant differences.

**Table 5 healthcare-08-00290-t005:** Bivariate logistic analysis of nutritional status and independent variables of ART recipients.

Variables	Nutritional Status	Odds Ratio	*p*-Value	95%CI
ART duration (≥2 year vs. <2 year)	Underweight	1.57	0.113	0.89–2.77
	Overweight	0.81	0.365	0.52–1.27
	Obesity	3.13	0.0001 *	1.72–5.71
Gender (males vs. females)	Underweight	1.71	0.066	0.96–3.03
	Overweight	0.80	0.318	0.51–1.23
	Obesity	0.23	0.0001 *	0.11–0.47
Age category (≥35 vs. <35)	Underweight	1.22	0.480	0.69–2.15
	Overweight	1.55	0.050 *	1.00–2.41
	Obesity	1.71	0.068	0.96–3.04
Employed vs. unemployed	Underweight	0.57	0.076	0.31–1.06
	Overweight	0.85	0.497	0.54–1.33
	Obesity	0.49	0.026 *	0.26–0.91
Income (low vs. high)	Underweight	0.58	0.087	0.31–1.01
	Overweight	0.84	0.455	0.54–1.31
	Obesity	0.49	0.030 *	0.26–0.93
Household size (1–4 vs. ≥5 members)	Underweight	1.38	0.279	0.76–2.49
	Overweight	1.59	0.045 *	1.01–2.53
	Obesity	0.70	0.231	0.40–1.24
Household income (low vs. high)	Underweight	0.42	0.003 *	0.23–0.74
	Overweight	0.90	0.688	0.57–1.44
	Obesity	0.76	0.364	0.42–1.36

* Indicates significant differences.

**Table 6 healthcare-08-00290-t006:** Bivariate logistic regression of age and nutritional status.

Variables	Categories	Odds Ratio	*p*-Value	95% CI
Waist circumference	Normal	Reference		
	Abdominal obesity	0.95	0.842	0.61–1.49
Waist/hip ratio (WHR)	Normal	Reference		
	Abdominal obesity	0.82	0.354	0.55–1.23
Waist-to-height ratio (WHtR)	Normal	Reference		
	Abdominal obesity	0.80	0.389	0.49–1.31
Nutritional status	Normal weight	Reference		
	Underweight	1.17	0.575	0.66–2.08
	Overweight/obesity	1.61	0.016 *	1.09–2.39

* Indicates significant differences. The independent factors significantly associated with nutritional status among the participants at multivariate logistic regression were age and gender, after controlling for ART duration, employment status, household size, and income. Those ≥35 years were 1.6 times more likely to be overweight (adjusted odds ratio (AOR) = 1.66; 95% CI: 1.13–244) than those >35 years. Males were less likely to develop abdominal obesity (AOR = 0.39; 95% CI: 0.26–0.58) than females.

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
