# Peer review of "The Nutritional Status of Adult Antiretroviral Therapy Recipients with a Recent HIV Diagnosis; A Cross-Sectional Study in Primary Health Facilities in Gauteng, South Africa"

_healthcare, 2020, doi:10.3390/healthcare8030290_

Round 1
Reviewer 1 Report
Based on 480 HIV-positive participants (age 18-49 years) receiving antiretroviral therapy (ART) from Gauteng Province, South Africa, the nutritional status defined by BMI and the associations between the duration of ART and nutritional status were investigated. The study is a cross-sectional one.
The authors concluded that there is a double burden of over- and under-nutrition among ART users. Thus, interventions to prevent overweight/obesity and to treat undernutrition should be integrated in routine HIV care.
I would make the following comments:
The introduction is somewhat long and should be shortened. The authors should focus on nutritional status in PLHIV and ART and streamline the introduction in this regard.
In the Material and Methods section the description of the sample size calculation is somewhat confusing and should be rewritten. Also, the data analysis section should be revised. The authors should describe the stratified analysis and the multivariable analysis in more detail. In addition, the confounder variables used in the final multivariable logistic regression models should be given. Also, the authors should clearly state (as in the abstract), how undernutrition and normal nutritional status due to BMI was defined.
Results: on page 4, lines 151/152: This sentence is very confusing and unclear. Furthermore, page 4, lines 161 and 162: ORs are given for males in comparison with females. It is not clear, whether these are results from bivariate logistic regression or multivariable regression? Page 5, line 169: this p-value of 0.204 belongs to the comparison regarding the WHtR-measure only.
Page 6, lines 183-192: the OR=0.42 for household income is not given in the table 5. Please check. The factors independently associated with nutritional status and the analyses stratified by age should be shown in a Table. What is meant by the term “AOR”? This should be also clarified in the statistical analysis section.
Table 5: are the ORs given in the table shown for men or women, employed vs. unemployed, age >= 35 vs. <35, etc.? Please clarify.
Table 2: Please add also height. BMI is given as median and interquartile range? This is not mentioned in the statistical analysis section (there is talk of mean values).
The discussion is somewhat redundant. I suggest to revise and streamline it.
Limitations section: the authors mention that selection bias could be an issue in the study. How could a selection have influenced the results? Furthermore, most likely, the findings of the study are not generalizable to other age-groups.
The manuscript should be edited regarding language and style.
Author Response
Dear Editor,
The corrections are written in blue. Kindly note that the line numbers have changed due to shortening the introduction.
Reviewer 1
The authors concluded that there is a double burden of over- and under-nutrition among ART users. Thus, interventions to prevent overweight/obesity and to treat undernutrition should be integrated in routine HIV care.
I would make the following comments:
- The introduction is somewhat long and should be shortened. The authors should focus on nutritional status in PLHIV and ART and streamline the introduction in this regard.
Response; the introduction has been revised and shortened. We removed the information that belonged to the general population, lines 26-65.
- In the Material and Methods section the description of the sample size calculation is somewhat confusing and should be rewritten. Also, the data analysis section should be revised. The authors should describe the stratified analysis and the multivariable analysis in more detail.
- In addition, the confounder variables used in the final multivariable logistic regression models should be given.
Response; the confounders have been included, lines 189-190.
- Also, the authors should clearly state (as in the abstract), how undernutrition and normal nutritional status due to BMI was defined.
Response; Undernutrition is indicated as BMI<18.5 kg/m2, normal BMI was 18.5–24.9 kg/m2. , line 111.
III. Results: on page 4, lines 151/152: This sentence is very confusing and unclear.
Response; the lines have been revised, lines 149-150.
- Furthermore, page 4, lines 161 and 162: ORs are given for males in comparison with females. It is not clear, whether these are results from bivariate logistic regression or multivariable regression?
Response; A bivariate logistic regression has been added, line 158.
- Page 5, line 169: this p-value of 0.204 belongs to the comparison regarding the WHtR-measure only.
Response; the P-value has been removed, line 167,
- Page 6, lines 183-192: the OR=0.42 for household income is not given in the table 5.
Response; we have added the OR =0.42 for the household income in table 5, line 193.
VII. The factors independently associated with nutritional status and the analyses stratified by age should be shown in a Table.
Response; table 6 has been added, line 198.
VIII. What is meant by the term “AOR”? This should be also clarified in the statistical analysis section.
Response; AOR meaning added in the statistical analysis
- Table 5: are the ORs given in the table shown for men or women, employed vs. unemployed, age >= 35 vs. <35, etc.? Please clarify.
Response; Categories have been added, line 193.
- Table 2: Please add also height. BMI is given as median and interquartile range? This is not mentioned in the statistical analysis section (there is talk of mean values).
Response, we have added height, and the median and interquartile range of height, WC, WHR, WHtR. We are now talking about medians, lines 148-161.
- The discussion is somewhat redundant. I suggest to revise and streamline it.
Response, we read the discussion repeatedly. We decided to delete information on the general population and left information that compare out findings with studies conducted among PLHIV and on ART, lines 199 -298.
Limitations section: the authors mention that selection bias could be an issue in the study. How could a selection have influenced the results? Furthermore, most likely, the findings of the study are not generalizable to other age-groups.
Response; we explained how we mitigated the potential of the convenience sampling technique to influence the results. We stated that data were collected over a long period and using different times. In addition, the time for fieldwork varied so that patients who collect their ART late in the day can be included in the sample, line 277.
The manuscript should be edited regarding language and style.
Response; the manuscript has been taken for language editing.
Thank you,
Reviewer 2 Report
Healthcare_review July 15, 2020
This review is for: “The nutritional status of adult antiretroviral therapy 2 recipients with a recent HIV diagnosis; a cross-3 sectional study in primary health facilities in 4 Gauteng Province, South Africa” by Mahlangu et. al. which contributes to the current literature in PLHIV research.
Of note:
Line 15 of abstract: suggested “ART recipients (N=480)…”
Line 73, perhap define “Universal Test and Treat”, is this a common term for this population? This may not be known to none content experts.
Line 87: there a shading at the end of the sentence.
Line 90: how do you define critically ill patients, please specify.
Line 94: gives should read gave, as the rest of the paragraph is written in the past-tense.
Line 95-96: the sentence “the facilities were offering ART refill…” I am not sure how to read this sentence, please specify (facilities in a square-mile radius, in that specific region, which facilities?). This being said the following few lines (96-98) do not entirely flow, consider revising.
Data collection instruments: researcher-designed, is this something that could be included in the supplemental? Would provide reader a bit more context.
Line 113: is the same scale used throughout the course of the study, same brand/model, and do you know what the margin of error on the scale is. Report here.
Line 132: what is a “forward stepwise procedure” how/why is this relevant.
Line 137: “informed consent in their language of preference…” was the consent translated. Perhaps explain what this is.
Line 173: the sub-division of <2 years and >2 years, was this based on the literature?
Line 181: Table 4, can you comment on some specific interventions that might assist with this issue around being overweight, i.e. exercise regimen, or consultation with a nutritionist, stress management, counseling…ect....
Line 191: Males- needs and “s”.
Line 230: Consider new paragraph, “the study further found that household income…” this seems to be a different topic.
Generally the discussion could benefit from a simple visual schematic, representing arrows with minimal text of weight trends, 1) vertical down arrow for undernutrition before at start of treatment, 2) horizontal arrow while on ART, 3) and a vertical up arrow for overnutrition/obesity.
Line 266-268: could this be incorporated and extrapolated into a potential recommendation or intervention, that ART patients live with friends or adult foster care, roommate. Is this linked to patients on ART being stressed and perhaps stress-eating, or binge-eating, is future work going to tie this into dietary recall or dietary journaling?
I would indicate at the end of the introduction, “this article is intended for general clinicians, nutritionists, dietitians, treating teams of ART, HIV researchers, policy development, psychosocial interventionists, social workers, ect…” this helps to guide and focus your readership.
Line 279: weight of the patient at enrolment, seems like a very key point, is this something that needs to be included in future studies.
Perhaps authors could make a general statement in the discussion around how long patients tend to be on ART, indefinitely? This being said, does the weight gain plateau after 2 yrs, or does it continue to rise…a bit more context around this would be helpful.
Author Response
Dear Editor,
Please find the corrections.
Reviewer 2
This review is for: “The nutritional status of adult antiretroviral therapy 2 recipients with a recent HIV diagnosis; a cross-3 sectional study in primary health facilities in 4 Gauteng Province, South Africa” by Mahlangu et. al. which contributes to the current literature in PLHIV research.
Of note:
- Line 15 of abstract: suggested “ART recipients (N=480)…”
Response; ART recipients (N=480) has been added in the abstract, line 15
- Line 73, perhap define “Universal Test and Treat”, is this a common term for this population? This may not be known to none content experts.
Response; Universal Test and Treat has been explained from the South African context, lines 61 - 63.
III. Line 87: there a shading at the end of the sentence.
Response; we deleted the shading, line 78.
- Line 90: how do you define critically ill patients, please specify.
Response; we revised critically ill and added “those who had obvious symptoms of opportunistic infection or had cognitive impairment due to HIV/AIDS infection or sub-mentality at the time of the study”, Lines 81 – 83.
- Line 94: gives should read gave, as the rest of the paragraph is written in the past-tense.
Response; “give” has been changed to “gave”, line 86.
- Line 95-96: the sentence “the facilities were offering ART refill…” I am not sure how to read this sentence, please specify (facilities in a square-mile radius, in that specific region, which facilities?). This being said the following few lines (96-98) do not entirely flow, consider revising.
Response; we revised the lines to “The selected facilities used in this study have been offering ART refill and follow-up on an appointment basis for PLHIV. The appointment schedule list was used to identify and select”, lines 88-90.
VII. Data collection instruments: researcher-designed, is this something that could be included in the supplemental? Would provide reader a bit more context.
Response; the instrument with relevant questions applicable to the manuscript has been included in the supplemental.
VIII. Line 113: is the same scale used throughout the course of the study, same brand/model, and do you know what the margin of error on the scale is. Report here.
Response; we used “a D-quip smart scale throughout the life course of the project, with a margin error between 0.1% - 0.2% of the actual weight”, lines 106 -107.
- Line 132: what is a “forward stepwise procedure” how/why is this relevant.
Response; forward stepwise regression adds variables one by one into the final model to understand the contribution of the previous variables now that another variable has been added. Both the backward and frontward stepwise procedure are used to find the influence of potential confounders and statistical significance on the dependant variables.
- Line 137: “informed consent in their language of preference…” was the consent translated. Perhaps explain what this is.
Response; the questionnaire and informed consent form had been translated into IsiZulu and Setswana; which are the local spoken languages, line 133.
- Line 173: the sub-division of <2 years and >2 years, was this based on the literature?
Response: this was not based on literature, but the duration on ART was used as a measure to assess whether duration on ART influences the nutritional status of PLHIV.
XII. Line 181: Table 4, can you comment on some specific interventions that might assist with this issue around being overweight, i.e. exercise regimen, or consultation with a nutritionist, stress management, counseling…ect....
Response; We apologize that we could not respond to this comment because we did not
understand what the reviewer meant.
XIII. Line 191: Males- needs and “s”-
Response; corrected, line 191.
XIV. Line 230: Consider new paragraph, “the study further found that household income…” this seems to be a different topic.
Response; revised, lines 230-236.
- Generally the discussion could benefit from a simple visual schematic, representing arrows with minimal text of weight trends, 1) vertical down arrow for undernutrition before at start of treatment, 2) horizontal arrow while on ART, 3) and a vertical up arrow for overnutrition/obesity.
Response; we apologize that we do not know what the reviewer is suggesting and we could not address this comment
XVI. Line 266-268: could this be incorporated and extrapolated into a potential recommendation or intervention, that ART patients live with friends or adult foster care, roommate. Is this linked to patients on ART being stressed and perhaps stress-eating, or binge-eating, is future work going to tie this into dietary recall or dietary journaling?
Response; literature and our personal experience of the communities were the study was conducted do not support this observation. In large households, more people are unemployed and are more likely to be food insecure when compared to those in smaller households.
XVII. I would indicate at the end of the introduction, “this article is intended for general clinicians, nutritionists, dietitians, treating teams of ART, HIV researchers, policy development, psychosocial interventionists, social workers, ect…” this helps to guide and focus your readership.
Response: we added the text in this regard in the introduction, lines 63-65.
XVIII. Line 279: weight of the patient at enrolment, seems like a very key point, is this something that needs to be included in future studies.
Response; we had stated in the limitations that we could not collect or analyse the weight of the patient at enrolment (because of the quantity of missing and incomplete data) in order to draw comparisons and assess the trend in weight gain during ART care, lines 282 – 285.
XIX. Perhaps authors could make a general statement in the discussion around how long patients tend to be on ART, indefinitely? This being said, does the weight gain plateau after 2 yrs, or does it continue to rise…a bit more context around this would be helpful.
Response; in line 270 and 273 we had stated that in cohorts of PLHIV in the US, Canada, and Brazil, the prevalence and incidence of obesity after ART initiation continued to increase, 22% of individuals with normal BMI became overweight and 18% of overweight individuals became obese within 3 years after ART initiation.
Thank you,
Round 2
Reviewer 1 Report
After Revision, the manuscript has improved. However, I have some further comments:
Table 6: it becomes not clear for the Reader that an age-stratified Analysis is presented in Table 6. It is not clear, what variables are included in the multivariate Analysis. The authors should Mention the variables in the text and as a footnote under Table 6. Altogether, the multivariate Analysis is difficult to understand as it is described by the authors. Please give more Details in the Data Analysis section to make it clear for the Reader how the multivariate model was conducted and stratified.
Did the authors Conduct a formal test on interaction with age? If yes, was the interaction significant?
page 3, line 126: please check....p-value > 0.20
page 4, line 155: do you mean ...p= <0.0001
Table 5: p-values of 0.000 are uncommon. It would be better to give p-values as p= <0.0001
page 6, lines 182/183: in Table 5 bivariate logistic Analysis results are shown, not multivariate logistic Regression results.
The English language style has improved, but there are still moderate English changes are necessary.
Author Response
Dear Reviewer,
Table 6: it becomes not clear for the Reader that an age-stratified Analysis is presented in Table 6. It is not clear, what variables are included in the multivariate Analysis. The authors should Mention the variables in the text and as a footnote under Table 6.
Response: The table show bivariate analysis and not multivariate analysis. We added the table in response to the reviewer comments, “The factors independently associated with nutritional status and the analyses stratified by age should be shown in a Table”. Since we conducted bivariate analysis, we included only the variables that define the nutritional status as presented in table 5.
Altogether, the multivariate Analysis is difficult to understand as it is described by the authors. Please give more Details in the Data Analysis section to make it clear for the Reader how the multivariate model was conducted and stratified.
Response: We had stated that all categorical variables associated with the outcome variables that had a p value < 0.20 in the bivariate analysis were entered in the multivariate logistic regression model using a forward stepwise regression.
Did the authors Conduct a formal test on interaction with age? If yes, was the interaction significant?
Response: While interaction effects are common in regression analysis, we did not conduct a forma interaction with age because we did not hypothesize that the effect of age the outcome variables is dependent on other variables.
page 3, line 126: please check; p-value > 0.20
Response: P-value has been changed to be < 0.20
page 4, line 155: do you mean ...p= <0.0001
Response: we have rephrased to P ≤ 0.0001
Table 5: p-values of 0.000 are uncommon. It would be better to give p-values as p= <0.0001
Response: P-values have been changed to ≤0.0001
page 6, lines 182/183: in Table 5 bivariate logistic Analysis results are shown, not multivariate logistic Regression results.
Response: we have removed multivariate. It is bivariate logistic analysis.
The English language style has improved, but there are still moderate English changes are necessary.
Thank you,